# On the Efficiency of Structured Pruning in Small Language Model Pretraining

## Abstract

Recent advancements in generative language models have intensified the need for efficient and deployable models within limited inference budgets, while companies possess enormous computational resources for training. This scenario opens a new regime and presents a fundamental question: given sufficient training resources but strict inference constraints, what is the most effective approach to obtain the best possible small generative language model? One solution is to utilize structured pruning to compress a large model to a small model. However, while structured pruning has shown promise compared to training a target-size model from scratch as shown in existing works, the overall efficiency becomes unclear when incorporating the cost of pretraining the large model that serves only as an intermediate step in our new scenario. In this paper, we first study the question of whether it is worth pretraining the large model even if it is never deployed. Our results show that once the pretraining cost of the large model is taken into account, existing pruning methods are less token-efficient than training the target-size model from scratch. Therefore, we further investigate how to improve the efficiency of the entire pipeline for producing small models. To this end, we propose an integrated enlarge-and-prune pipeline, which combines enlarged model training, pruning, and recovery under a single cosine annealing learning rate schedule. This approach is further complemented by an iterative structured pruning method for the gradual removal of parameters. We conduct comprehensive experiments on compressing 2.8B models to 1.3B with up to 2T tokens in pretraining. Our results demonstrate that the integrated approach not only provides insights into the token efficiency of structured pruning but also achieves superior performance of pruned models.

## 1 Introduction

The deployment of generative language models on resource-constrained devices has become increasingly critical for modern applications. Tech companies face growing demands to deploy generative language models on smartphones and edge devices (Abdin et al., 2024; Gunter et al., 2024). Given the severe memory limitations of these devices, model sizes must remain small, typically under 3 billion parameters for practical deployment. Meanwhile, these companies possess enormous computational resources and are willing to invest substantially in training costs to achieve optimal small generative language model performance. This scenario opens a new regime and presents a fundamental question for practitioners: given sufficient training resources but strict deployment size constraints, what is the most effective approach to obtain the best possible small generative language model from scratch?

The conventional approach involves scaling up training tokens—simply training the small model longer (Kaplan et al., 2020; Hoffmann et al., 2022). However, the performance of the small model under this approach quickly plateaus, yielding diminishing returns despite additional training. Recently, structured pruning pipelines (Xia et al., 2022; Wang et al., 2019; Kwon et al., 2022) offer a promising approach through a two-stage process of pruning and recovery. The pruning stage compresses an existing large model to the target small size through parameter removal, followed by a recovery stage that restores the model's performance through continual pretraining. In practice, the pruned model achieves superior performance while requiring fewer training tokens than training a small model from scratch. As a result, the two-stage pruning pipeline is more token efficient than

conventional training a small model from scratch (Xia et al., 2023; Muralidharan et al., 2024). This efficiency stems from the pruning-informed initialization partially inherited and retained from the large model.

Although two-stage pruning pipelines have demonstrated strong token efficiency, their conclusions cannot be directly applied to our new scenario. First, prior studies often assume access to a pretrained large model and thus ignore the cost of training it. In contrast, our goal is to build small models entirely from scratch. In this case, the large model only serves as an intermediate step and is never deployed. Therefore, its pretraining cost must be included in the overall token budget. Second, many existing works (Xia et al., 2023; Muralidharan et al., 2024) train the pruned model on a dataset different from the one used for small-model-from-scratch baselines. These dataset mismatches make it unclear whether observed improvements stem from the pruning itself or simply from more favorable training data. Consequently, it is unclear whether pruning still offers token efficiency advantages over training a small model from scratch when all training costs and datasets are properly aligned.

To study this question, we extend conventional two-stage pruning by explicitly incorporating the pretraining of the large model, which we term the *naive enlarge-and-prune pipeline*. This pipeline enables, for the first time, a rigorous study of the end-to-end token efficiency of pruning in producing small models from scratch. We conduct pilot experiments to compare the naive enlarge-and-prune pipeline against simply training small models from scratch given the same training tokens on the same dataset. Figure 1a shows that the naive pipeline fails to consistently achieve superior token efficiency compared to direct pretraining of small models. This inefficiency can be attributed primarily to the divided nature of the naive pipeline, where the training processes across successive stages often lack optimal alignment. For example, the initial learning rate of each new stage substantially exceeds the final learning rate of its preceding stage, as illustrated in Figure 2a and Figure 2b. This discontinuity triggers a sharp increase in the loss curve, as shown in Figure 1b, leading to catastrophic forgetting previously acquired knowledge and ultimately degraded model performance. Furthermore, the naive pipeline employs disparate training objectives between stages, such as auxiliary regularization loss for mask learning (Xia et al., 2023) and activation-based proxy losses (Muralidharan et al., 2024), causing suboptimal initialization for the recovery stage.

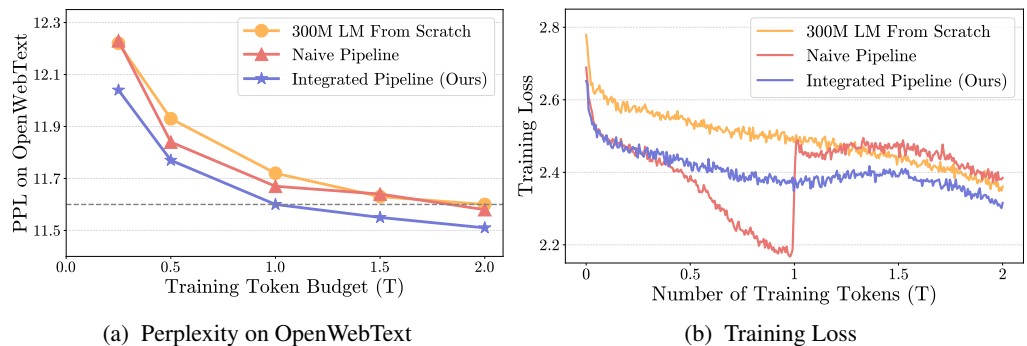

(a) Perplexity on OpenWebText

(b) Training Loss

Figure 1: *Naive pipeline*: 1T for enlarged model training and 1T for pruned model recovery, each with separate learning rate decay. *Integrated pipeline*: single learning rate schedule matching target-size model training, pruning started at 1T. *Left*: Perplexity on OpenWebText of 300M model trained with multiple training token budgets from 0.25T to 2T. The integrated pipeline is **2x** token efficient than the naive pipeline. *Right*: Training loss curves for enlarge-and-prune pipelines. Training 300M models with 2T total token budget.

To overcome the aforementioned limitations of the naive enlarge-and-prune pipeline, we propose IDEA Prune, an **I**ntegrate**D E**nlarge-**A**nd-**Prune** pipeline with a novel iterative structured pruning specifically for producing small language models. The integrated pipeline organically combines the enlarged model pretraining, pruning, and pruned model recovery stages under one cosine annealing learning rate schedule, as illustrated in Figure 2c. This helps to mitigate the knowledge loss caused by the rising learning rate in naive pipelines (see Figure 1b), where each stage uses an individual cosine annealing with warm up (Muralidharan et al., 2024; Xia et al., 2023). Furthermore, we extend iterative pruning (Zhu & Gupta, 2017; Louizos et al., 2017) to structured Feed-Forward s (FFNs) compression. Our approach progressively removes neurons, which corresponds to the rows

of weight matrices with the same indices. We identify the surviving neurons based on element-wise importance scores computed across FFN weight matrices (Molchanov et al., 2019; Zhang et al., 2022; Ding et al., 2019). With iterative width reduction and parameter updates, our approach effectively redistributes the model capacity among surviving neurons, facilitating smooth compression and enhanced performance.

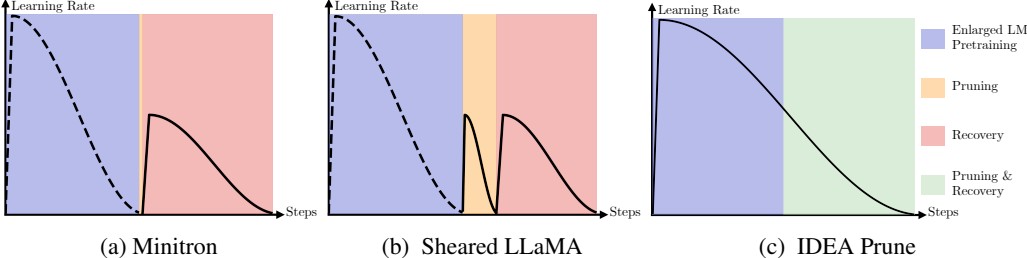

(a) Minitron  (b) Sheared LLaMA  (c) IDEA Prune

Figure 2: Learning rate schedule for separate (Figure 2a, 2b) and integrated (Figure 2c) enlarge-and-prune pipelines. Minitron uses single shot pruning, resulting in a narrow pruning stage. Our integrated pipeline uses one single learning rate decay schedule for all stages. We use iterative structured pruning to further integrate pruning and recovery.

We demonstrate the effectiveness of IDEA Prune through extensive experiments by compressing a 2.8B model to 1.3B parameters with up to 2T training tokens. IDEA Prune extends pruning beyond continual pretraining to the full pretraining regime, further unleashing the power of generative small language models. In controlled comparisons with existing approaches—one-shot random pruning, learned mask pruning (Xia et al., 2023), and activation-based pruning (Muralidharan et al., 2024)—our method demonstrates consistent performance improvements across multiple benchmarks. Notably, IDEA Prune significantly improves MMLU accuracy to 46.4%, compared to 31.4-33.4% for baseline methods. Our analysis further reveals two key insights: (1) intermediate checkpoints, though weaker than final checkpoints in absolute performance, provide better starting points for pruning; and (2) there exists an optimal enlarged model size—approximately 2.6× the target size—that yields the best pruned models. Together, these findings also offer practical guidance for selecting large models under conventional pruning settings. Finally, for rigorous evaluation of the pruning methodology itself, we isolate our analysis from complementary techniques such as knowledge distillation, though we conduct separate ablation studies to confirm knowledge distillation can be effectively combined with our method for further improvements.

## 2 BACKGROUND AND RELATED WORK

### 2.1 ITERATIVE PRUNING

Iterative pruning is a parameter reduction technique that alternates between parameter optimization and selective removal (Zhu & Gupta, 2017; Louizos et al., 2017). During pruning, the method computes importance scores for parameters and gradually removes those less critical. For a model with parameters $\boldsymbol{\theta}^{(t)} \in \mathbb{R}^N$, we measure parameter importance by sensitivity (Molchanov et al., 2019), $\boldsymbol{\omega}^{(t)} \in \mathbb{R}^N$, which represents the first-order approximation of each parameter's impact on the training loss. See Appendix C.1 for the definition. We then generate a binary mask $\boldsymbol{v}^{(t)}$ based on the ranking of $\boldsymbol{\omega}^{(t)}$, and apply it to obtain the pruned parameters: $\boldsymbol{\theta}^{(t+1)} = \boldsymbol{v}^{(t)} \odot \boldsymbol{\theta}^{(t)}$. This progressive approach enables a smooth transition from larger to smaller models. Note that these works focus on unstructured pruning, while we extend iterative pruning to structured pruning of FFN width in Section 3.2.

### 2.2 STRUCTURED PRUNING OF FFN

Transformers (Vaswani, 2017), the mainstream architectures in current generative language models (Radford et al., 2019; Jiang et al., 2023; Dubey et al., 2024; Gunter et al., 2024), consist of sequential layers containing attention and Feed-Forward Network (FFN) components. We consider an FFN layer equipped with a gated Sigmoid Linear Unit (SiLU) activation function, denoted as $\sigma(\cdot)$. Given

the input $\boldsymbol{x} \in \mathbb{R}^d$, the layer's output $\boldsymbol{y} \in \mathbb{R}^d$ is defined as

$$\boldsymbol{y} = \boldsymbol{W}_{\text{down}}^{\top}(\sigma(\boldsymbol{W}_{\text{up}}\boldsymbol{x}) \odot (\boldsymbol{W}_{\text{gate}}\boldsymbol{x}))$$

$$= \sum_{i=1}^{h} \boldsymbol{W}_{\text{down}[i,:]}^{\top} \left( (\boldsymbol{W}_{\text{up}[i,:]}x)(\boldsymbol{W}_{\text{gate}[i,:]}\boldsymbol{x}) \right),$$

where $\boldsymbol{W}_{\text{up}}, \boldsymbol{W}_{\text{gate}}, \boldsymbol{W}_{\text{down}} \in \mathbb{R}^{h \times d}$, $d$ is the input dimension, $h$ is the hidden dimension or the number of neurons, and $\odot$ represents element-wise multiplication. Structured pruning of FFN width targets the removal of neurons, which corresponds to the rows in $\boldsymbol{W}_{\text{up}}$, $\boldsymbol{W}_{\text{gate}}$, and $\boldsymbol{W}_{\text{down}}$ with the same indices, for example, $\boldsymbol{W}_{\text{up}[i,:]}, \boldsymbol{W}_{\text{gate}[i,:]}, \boldsymbol{W}_{\text{down}[i,:]}$. This is achieved by finding a binary mask $\boldsymbol{m} \in \{0,1\}^h$ that determines which neurons to retain. The pruned weight matrices are:

$$\boldsymbol{W}_{\text{up}}^{'} = \text{diag}(\boldsymbol{m}) \times \boldsymbol{W}_{\text{up}}, \boldsymbol{W}_{\text{gate}}^{'} = \text{diag}(\boldsymbol{m}) \times \boldsymbol{W}_{\text{gate}}, \boldsymbol{W}_{\text{down}}^{'} = \text{diag}(\boldsymbol{m}) \times \boldsymbol{W}_{\text{down}}. \quad (2.1)$$

After mask application, the zero columns in each matrix are eliminated to obtain compressed weight matrices, resulting in reduced memory usage and lower inference latency.

In this paper, we focus on pruning FFN width. We do not prune the depth as Muralidharan et al. (2024) suggest pruning width (i.e., the hidden dimension of FFN and the number of attention heads) is more efficient than pruning depth (i.e., the number of transformer layers). Moreover, we exclude attention layers from our pruning, both for simplicity and due to their relatively minor contribution to the total parameters (less than 20%).

## 2.3 NAIVE ENLARGE-AND-PRUNE PIPELINE

A naive enlarge-and-prune pipeline (Xia et al., 2023; Muralidharan et al., 2024) consists of three distinct stages: enlarged model pretraining, pruning, and pruned model recovery. From the perspective of learning rate specifically, each of the three stages uses its own independent learning rate schedule (Xia et al., 2023). The learning rate at the $t$-th step through the naive enlarge-and-prune pipeline is

$$\beta^{(t)} = \begin{cases} \eta(t; T_l, \eta_1, \eta_2) & \text{if } 0 < t \leq T_l, \\ \eta(t - T_l; T_p, \eta_3, \eta_4) & \text{if } T_l < t \leq T_l + T_p, \\ \eta(t - T_l - T_p; T_r, \eta_5, \eta_6) & \text{if } T_l + T_p < t \leq T, \end{cases}$$

where $\eta(\cdot; \cdot)$ is the cosine annealing learning rate schedule (Loshchilov & Hutter, 2016) (see Appendix B.2), $T$ is the total training steps and $T_l, T_p, T_r$ are the number of enlarged model training steps, pruning steps, pruned model recovery steps, respectively. An example can be found in Figure 2b. In the naive enlarge-and-prune pipeline, we need to tune different sets of peak and end learning rates $\eta_1, ..., \eta_6$, which requires considerable effort to find the optimal learning rate sets.

## 2.4 TRAIN LARGE, THEN COMPRESS

Li et al. (2020) demonstrate that pretraining larger models with early stopping is computationally more efficient than training smaller models to convergence. To meet test-time constraints, they compress the large model through downstream task adaptation, which outperforms naive pretraining and finetuning of equivalent-sized models.

While this approach shares similarities with our method, there are several key distinctions. Our work employs structured pruning, which maintains inference speeds, whereas their unstructured pruning approach fails to match the latency of similarly-sized dense models due to limited hardware support for sparse matrix operations. Moreover, we conduct pruning exclusively during pretraining to produce a general-purpose pretrained model, in contrast to their task-specific pruning during finetuning, which leaves the model's broader applicability unclear. Last, we focus on decoder-only models, while their findings are based on encoder-only models like RoBERTa (Liu, 2019)—a significant architectural difference given the disparate pruning behaviors observed in recent works (Xia et al., 2022; 2023).

## 2.5 STRUCTURED PRUNING IN POST-TRAINING

Recent works (Ma et al., 2023; Ashkboos et al., 2024; An et al., 2024; Wei et al., 2024) succeed in compressing pretrained generative models through structured pruning. These works focus on how

to obtain pruned models with better performance when pretrained models are off-the-shelf, while we are interested in how to optimize the entire enlarge-and-prune pipeline. Although the goals are different, their pruning methods can be adopted into our integrated pipeline, as we adopt Minitron and Sheared LLaMA in Appendix H.2 and show improvement over the naive pipelines. However, the efficacy of these post-training pruning methods is unclear in pretraining, and there may be intrinsic problems during the adoption. Therefore, we leave the adoption of post-training pruning to our pipeline as future work.

## 3 METHOD

We propose IDEA Prune, an **IntegrateD Enlarge-And-Prune** pipeline with a novel iterative structured pruning for producing small generative language models from scratch. IDEA Prune combines the enlarged model pretraining, pruning, and pruned model recovery into a single training run. It mitigates the performance degradation of the naive enlarge-and-prune pipeline and reduces the cost of learning rate tuning. Moreover, we unify the pruning and recovery stages by iteratively pruning the neurons based on their importance scores, which provides an accurate and smooth pruning of redundant neurons, preventing drastic performance drop.

### 3.1 INTEGRATED ENLARGE-AND-PRUNE PIPELINE

We denote the number of enlarged model pretraining, pruning, and pruned model recovery steps as $T_l, T_p, T_r$, respectively. The learning rate for weight matrix update at step $t$ is given by

$$\alpha^{(t)} = \eta(t; T_l + T_p + T_r, \eta_p, \eta_e),$$

where $\eta(\cdot; \cdot)$ is the cosine annealing learning rate schedule defined in (B.2). Unlike the naive enlarge-and-prune pipeline in Section 2.3, our integrated approach eliminates learning rate warm-up during pruning and recovery stages, thus mitigating potential knowledge loss.

Critically, this learning rate schedule method is not constrained to specific pruning techniques. we conduct ablation studies to evaluate its effectiveness across diverse approaches, including one-shot random pruning, activation-based pruning, and learned mask pruning in Section H.2.

Furthermore, the integrated pipeline offers additional flexibility by enabling its application to existing pretrained models with available intermediate checkpoints and historical learning rate schedules, as discussed in Section 5.1.

### 3.2 ITERATIVE STRUCTURED PRUNING IN FFN

To complement the integrated learning rate schedule, we propose a specialized iterative pruning method for FFN width based on the importance score of neurons. This approach further aligns the pruning and recovery stages by mitigating the suboptimal model initialization inherent in naive enlarge-and-prune pipelines during recovery training.

Concretely, for each element in a weight matrix $\boldsymbol{W}^{(t)} \in \mathbb{R}^{h \times d}$ at the $t$-th step, we define the importance score as the sensitivity $\boldsymbol{S}_{ij}^{(t)}$ defined in (C.1). Since the sensitivity is defined on the full dataset, we use the moving average of importance scores on a mini-batch of size $B$ to approximate it as

$$\widetilde{\boldsymbol{S}}_{ij}^{(t)} = (1 - \lambda)\boldsymbol{T}_{ij}^{(t)} + \lambda\widetilde{\boldsymbol{S}}_{ij}^{(t-1)}, \tag{3.1}$$

where

$$\boldsymbol{T}_{ij}^{(t)} = \left| \frac{1}{B} \sum_{n=1}^{B} \nabla l_n(\boldsymbol{W}_{ij}^{(t)})\boldsymbol{W}_{ij}^{(t)} \right|,$$

$\lambda$ is the moving average coefficient, and $l_n(\cdot)$ is the loss of the $n$-th sample in the mini-batch. The importance score for the $k$-th neuron is then derived by combining the moving average scores across the three weight matrices $\boldsymbol{W}_{\text{up}}, \boldsymbol{W}_{\text{gate}}, \boldsymbol{W}_{\text{down}}$:

$$\boldsymbol{c}_k^{(t)} = f_2\left(f_1(\widetilde{\boldsymbol{S}}_{\text{up}[k,:]}^{(t)}), f_1(\widetilde{\boldsymbol{S}}_{\text{gate}[k,:]}^{(t)}), f_1(\widetilde{\boldsymbol{S}}_{\text{down}[k,:]}^{(t)})\right), \tag{3.2}$$

where $f_1 : \mathbb{R}^d \to \mathbb{R}$ aggregates element-wise scores into a scalar and $f_2 : \mathbb{R}^3 \to \mathbb{R}$ combines across the three matrices. Options of $f_1$ and $f_2$ can be $\max(\cdot), \mathrm{mean}(\cdot)$.

Finally, we update the pruning mask using a scheduled sparsity by

$$\boldsymbol{m}_k^{(t)} = \begin{cases} 1 & \text{if } \boldsymbol{c}_k^{(t)} \text{ is in top } r(t; T_p) \text{ of } \boldsymbol{c}^{(t)}, \\ 0 & \text{otherwise,} \end{cases} \tag{3.3}$$

where $r(t; T_p)$ is a cubically decreasing function, defined in Appendix C.2, to reach the target sparsity. We provide a summary of the IDEA Prune pipeline in Appendix D.

## 4 EXPERIMENTS

### 4.1 EXPERIMENT SETUPS

**Model Architectures.** For baselines, we choose a 1.3B model as the target-size model, unless specified otherwise. The hidden dimension of its FFN layers is 6528. To design an enlarged model for pruning, we only increase the hidden dimension of the 1.3B model's FFN layer to $2048 \times 8$, resulting in a 2.8B model. This is because pruning width is more efficient, as we have stated in Section 2.2. See Appendix F for model configuration details.

**Datasets.** We train models on an open-source pretraining corpus, DCLM (Li et al., 2024), which contains 4T unique tokens of diverse domains.

**Baselines.** We compare against (i) training the 1.3B model from scratch with 1T and 2T tokens, and (ii) naive enlarge-and-prune pipelines using three pruning techniques:

• *One-shot random pruning (OSRP).* It randomly generates the pruning mask that satisfies the target sparsity in a single step.

• *Minitron.* Minitron uses the activation-based pruning. It computes the pruning mask based on the activation in FFN layers once. See Appendix E for the detailed computation. Note that we do not apply the distillation that is introduced in its original method.

• *Sheared LLaMA.* We obtain the pruning mask by learning, following the pruning method in Sheared LLaMA (Xia et al., 2023). This pruning method brings auxiliary parameters and changes the training objectives. The pruning process consumes 100B tokens. Note that we do not apply the dynamic batch loading that is introduced in its original method.

For the naive enlarge-and-prune baselines, the 2.8B model is pretrained for 1T tokens, pruned to 1.3B using one of the above techniques, and then continually pretrained for another 1T tokens.

Peak-end Learning rates follow optimal hyperparameter transfer (Yang et al., 2022): $(3.75 \times 10^{-3}, 1.875 \times 10^{-5})$ for 1.3B models (small model from scratch and pruned model recovery) and $(2.5 \times 10^{-3}, 1.25 \times 10^{-5})$ for 2.8B models.

**IDEA Prune Training.** We allocate 1T, 0.5T, and 0.5T tokens to the enlarged model training, pruning, recovery stages, respectively. We apply the one cosine annealing learning rate schedule through the entire training, with the peak-end learning rate of $(3.75 \times 10^{-3}, 1.875 \times 10^{-5})$.

**Hyperparameters.** We set the sequence length to 4096 and the batch size to 1024.

**Evaluations.** We report the perplexity on the test set of OpenWebText (OWT) (Gokaslan et al., 2019), the 0-shot accuracy on ARC-Challenge (ARC-C) (Clark et al., 2018) and HellaSwag (Zellers et al., 2019), the 1-shot accuracy on TriviaQA (Joshi et al., 2017), and the 5-shot accuracy on MMLU (Hendrycks et al., 2020).

### 4.2 ENLARGE-AND-PRUNE IN PRETRAINING

In this section, we investigate the effectiveness of IDEA Prune compared to the naive pipeline with baseline pruning methods. We also study the token efficiency of the enlarge-and-prune pipeline compared to training small models from scratch.

| Method | OpenWebText ↓ | Arc-C ↑ | Hellaswag ↑ | TriviaQA ↑ | MMLU ↑ |
|---|---|---|---|---|---|
| 2.8B-1T from scratch | 8.12 | 46.0 | 57.7 | 37.3 | 50.8 |
| 1.3B-1T from scratch | 9.10 | 39.3 | 52.8 | 30.3 | 28.9 |
| 1.3B-2T from scratch | 8.95 | **39.4** | 53.6 | 30.5 | 45.7 |
| OSRP (1.3B) | 8.98 | 38.9 | 53.4 | 29.6 | 32.5 |
| Minitron (1.3B) | 8.97 | 38.7 | 53.6 | 30.7 | 31.4 |
| Sheared LLaMA (1.3B) | 8.96 | **39.4** | 53.6 | 30.1 | 33.4 |
| IDEA Prune (1.3B) | **8.88** | 39.0 | **54.0** | **31.1** | **46.4** |

Table 1: Evaluation of models trained from scratch or through enlarge-and-prune pipelines. We train a 1.3B model for 2T tokens as the baseline. We apply naive pipelines with different pruning methods to prune a 2.8B model to 1.3B with 2T tokens. We report the perplexity(↓) of OpenWebText and accuracy(↑) of other benchmarks. The best results are in **bold**.

The experiment results are in Table 1. IDEA Prune shows improvement over the naive enlarge-and-prune pipeline with baseline pruning methods. The improvement on OpenWebText and MMLU is the most significant, as our method attains an OpenWebText perplexity of 8.88, outperforming the previous best of 8.96 achieved by Sheared LLaMA and significantly advances MMLU accuracy to 46.4%, compared to 31.4-33.4% for baseline methods. However, different pipelines show minimal difference on reading comprehension tasks: ARC-Challange, Hellaswag, and TriviaQA. This is likely because performance saturates on these tasks. As we increase the training tokens from 1T to 2T for the 1.3B model, the comprehension tasks are not significantly improved, but MMLU continues to improve.

The performance of the best pruning approaches (IDEA Prune) is slightly better than or comparable to the 1.3B-2T baseline as shown in Table 1. Some naive enlarge-and-prune pipelines are even worse than training from scratch. This implies that the enlarge-and-prune pipeline does not always increase token efficiency compared to training target-size models from scratch given the same training tokens, highlighting the need for careful pipeline selection.

It is important to note that the experiment setup of IDEA Prune is not fully optimized. In particular, the allocation of the token budget across stages and the choice of peak and end learning rates were not tuned for best performance. As discussed in Section H.1, an optimal allocation dedicates roughly 30% of the tokens to enlarged model training and the remaining 70% to pruning and recovery, but we use 50%-50% to align with naive pipelines for fair comparison. Moreover, optimal hyperparameters transfer may not work for IDEA Prune, since the model size evolves dynamically throughout training. Nevertheless, even under this suboptimal configuration, IDEA Prune achieves improvements over both training small models from scratch and naive pipelines, and further gains may be possible with careful tuning.

## 5  ANALYSIS

In this section, we ablate the pruning and recovery stages in the enlarge-and-prune pipeline, particularly, the initialization and learning rate schedule in Section 5.1. We analyze how the enlarged model size affects the final pruned model performance in Section 5.2. We defer the discussion of the robustness of the hyperparameters in Appendix H.1 and the extension of our proposed integrated pipeline to baseline pruning methods in Appendix H.2.

### 5.1  ANALYSIS OF INITIALIZATION AND LEARNING RATE

If we only consider the pruning and recovery stages, the major differences of the integrated and naive enlarge-and-prune pipelines are twofold: the *initial weights* of the model before pruning and the *learning rate schedule*.

**Initialization.** At the beginning of the pruning stage, naive enlarge-and-prune pipelines initialize the model by the last checkpoint of an enlarged model trained with, e.g., 1T tokens. In contrast, IDEA Prune loads the model weights that are equivalent to the intermediate checkpoint of an enlarged model with a larger token budget, e.g., 2T tokens if we did not prune the enlarged model and

| Model | LR Schedule | OpenWebText ↓ | Comp Avg ↑ | MMLU ↑ |
|---|---|---|---|---|
| 2.8B-2T@1T | - | 9.99 | 38.3 | 26.0 |
| 1.3B pruned from 2.8B-2T@1T | Resumed | **8.88** | **41.4** | **46.4** |
| | Restarted | 8.94 | 41.1 | 37.0 |
| 2.8B-1T@1T | - | 8.12 | 46.2 | 50.8 |
| 1.3B pruned from 2.8B-1T@1T | Resumed | 8.89 | 41.5 | 38.2 |
| | Restarted | 8.95 | 41.4 | 36.8 |

Table 2: Ablation of the initialization and the learning rate schedule in enlarge-and-prune pipelines. Our integrated enlarge-and-prune pipeline is equivalent to 2.8B-2T@1T initialization with a resumed learning rate schedule. Pruned model size: 1.3B.

continued to fully train the enlarged model. To ablate the impact of the initialization, we initialize the model by two checkpoints: the checkpoint of the 2.8B-2T model at the 1T-th step and the checkpoint of the 2.8B-1T at its last step.

**Learning rate schedule.** In IDEA Prune, we resume the learning rate schedule of the 2.8B-2T model at the 1T point, which provides a relatively small learning rate continuation, while naive enlarge-and-prune pipelines restart the cosine learning rate schedule with a linear warmup. We formalize these approaches into two learning rate schedule types: first, the *resumed* learning rate schedule, corresponding to the integrated enlarge-and-prune pipeline, mathematically represented by $\eta(t+T_l; T_l+T_p+T_r, \eta_p, \eta_e)$, where the learning rate continues from a previous training stage; and second, the *restarted* learning rate schedule, corresponding to the naive enlarge-and-prune pipeline, represented by $\eta(t; T_p+T_r, \eta_p, \eta_e)$, which initiates a new learning rate schedule from the beginning. We fix $\eta_p, \eta_e$ in each setting.

**Discussion 1.** We present the results in Table 2. First, both the resumed learning rate schedule and the use of intermediate checkpoints are crucial components; omitting either leads to substantial degradation in MMLU. Intriguingly, we find that the learning rate schedule has a greater impact than initialization, with the resumed schedule consistently outperforming the restarted approach across different initialization methods.

**Discussion 2.** Contrary to conventional wisdom, our results challenge the presumption that better initialization directly translates to better pruned model performance. Most notably in the resumed learning schedule, we discover that an intermediate checkpoint, despite showing lower performance than the final checkpoint, paradoxically provides a more favorable starting point for pruning, resulting in higher pruned model performance. This counterintuitive result stems from the learning dynamics: the final checkpoint's convergence at a low learning rate creates a mismatch with the resumed schedule's relatively higher learning rate, effectively mimicking a restarted schedule but with smaller peak learning rates. These findings emphasize the critical importance of aligning initialization with learning rate schedules in enlarge-and-prune pipelines.

## 5.2 ANALYSIS OF ENLARGED MODEL SIZE

Unlike pruning existing enlarged models, enlarge-and-prune pipelines have more freedom on the choice of the enlarged model size. Therefore, we investigate the impact of enlarged model size on performance through the enlarge-and-prune pipelines. Starting with a 300M parameter baseline model featuring an FFN layer with $3 \times 1024$ hidden dimensions (detailed architecture in Appendix F), we explore enlarged models by incrementally increasing FFN width to 4x, 6x, 8x, 12x, 16x of 1024, while keeping other model parameters constant. We conduct IDEA Prune using 500B tokens from DCLM, allocating 250B tokens for enlarged model training and the remaining 250B tokens for pruning and recovery.

The perplexity of the pruned model with 1.3x, 2x, 2.6x, 4x, and 5.3x enlargement factors is 11.86, 11.79, 11.77, 11.79, 11.84, respectively, while the 300M model trained from scratch baseline is 11.93. Notably, integrated pipelines across all enlarged model sizes significantly outperform the training-from-scratch baseline. This robust performance across a wide range of model sizes demonstrates the flexibility of our integrated enlarge-and-prune pipeline and little need for extensive model size tuning. Additionally, the results show an important trade-off between model capacity and pruning efficiency. Models with smaller FFN width enlargement factors (close to 1.3x) lack sufficient

capacity to learn rich representations, while extremely large models (approaching 5.3x) suffer from pruning degradation. Based on these observations, we identify 2.6x as the optimal FFN width enlargement factor.

## 6  COMBINATION OF KNOWLEDGE DISTILLATION

Knowledge distillation (KD) (Kim & Rush, 2016; Hinton, 2015) is another prominent approach for developing models with the help of enlarged models, where a small student model learns to mimic a large teacher model. Recent research (Team et al., 2024; Liang et al., 2023; Gunter et al., 2024; Li et al., 2023) finds pruning and distillation are complementary and often applies them together to obtain high performance models from existing pretrained models. As we have stated in Section 1, we are interested in the token efficiency of the entire process, including the training of the teacher model in KD. Thus, we study how distillation affects the enlarge-and-prune pipeline under a fixed training token budget for the enlarged (teacher) model pretraining, pruning, and pruned model recovery.

To start with, we train a 2.8B-2T model using 2T tokens as the teacher model. We establish a KD baseline by training a 1.3B model from scratch over 1T tokens with KD. In our enlarge-and-prune pipeline, we prune and train the 2.8B-2T@1T intermediate checkpoint using a resumed cosine learning rate schedule for 1T tokens—equivalent to the integrated pipeline with 2T tokens as shown in Section 5.1. During pruning and recovery, we use the 2.8B-2T model as the KD teacher.

As shown in Table 3, pruning with distillation outperforms the pruning baseline across most benchmarks and even surpasses the KD baseline on all benchmarks, particularly on OpenWebText and MMLU. These results reinforce our earlier conclusion in Section 4.2 and extend it to the KD regime: well-designed enlarge-and-prune pipelines, e.g., IDEA Prune, can exceed the performance of target-size models trained from scratch using KD.

| Method | OpenWebText | Arc-C | Hellaswag | TriviaQA | MMLU |
|---|---|---|---|---|---|
| Pruning w/o KD | 8.879 | 39.0 | **54.0** | 31.1 | 46.4 |
| 1.3B-1T w/ KD | 8.949 | 39.6 | 52.2 | 31.7 | 44.5 |
| Pruning w/ KD | **8.862** | **39.8** | 52.7 | **32.0** | **46.6** |

Table 3: Enlarge-and-prune pipeline with KD. Teacher model size: 2.8B-2T. The KD baseline: training a 1.3B model for 1T tokens with KD from scratch. *pruning w/ KD*: pruning (including recovery) the 2.8B-2T@1T checkpoint for 1T tokens with KD.

## 7  DISCUSSIONS

**Tokens or FLOPs?** In the comparison between the enlarge-and-prune pipeline and training small models from scratch, it is more practical to fix the number of training tokens than FLOPs. This is because current pretraining bottleneck is the availability of tokens rather than training resources, often represented by FLOPs, and our approach is designed specifically for this scenario, where practitioners have sufficient resources, i.e., FLOPs, and want to produce the best possible small models. Despite its practical justification, we also compare the enlarge-and-prune pipeline with small models from scratch under the same FLOPs in Appendix I. The results in Figure 4 show IDEA Prune pipeline outperforms the naive pipeline at all FLOPs budgets, and the small model from scratch baseline at higher FLOPs budgets, which is practical in our scenario of interest when practitioners have sufficient computational resources.

**Significance and Reliability.** First, while the performance gains of IDEA Prune over small models from scratch may appear modest, they are meaningful in the context of small language models, which tend to saturate quickly due to their limited capacity. Even incremental improvements in such a saturated regime represent a noteworthy achievement. Second, with respect to the potential noise in our results, we acknowledge that running multiple experiments with different random seeds would be ideal but is infeasible given the high cost of pretraining. Instead, we rely on prior findings that once models reach the convergence phase, performance metrics exhibit minimal variance across seeds (van der Wal et al., 2025). As shown in Table 7, final-stage results remain stable, indicating our models have reached convergence. Consequently, the final reported improvements are unlikely to result from random noise.

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

## A USE OF LARGE LANGUAGE MODELS

We use large language models as a tool to polish the paper, including but not limited to correcting grammar mistakes, reorganize existing content in an easier way to read, and adjusting the layout.

## B MODEL OPTIMIZATION

### B.1 ADAM OPTIMIZER

Adam optimization algorithm is popular in training large language models. Given a weight matrix $\boldsymbol{W}^{(t-1)}$ at the $t$-th step, we update the matrix $\boldsymbol{W}^{(t-1)}$ from the last step by

$$\boldsymbol{W}^{(t)} = \boldsymbol{W}^{(t-1)} - \gamma^{(t)}\widetilde{\boldsymbol{G}}^{(t)},$$

where $\gamma^{(t)}$ is the learning rate, which is often scheduled by a cosine annealing with a linear warm-up, and $\widetilde{\boldsymbol{G}}^{(t)}$ is calculated as

$$\widetilde{\boldsymbol{G}}^{(t)} = \frac{\widetilde{\boldsymbol{M}}^{(t)}}{\sqrt{\widetilde{\boldsymbol{V}}^{(t)}} + \epsilon},$$

where $\epsilon$ is a hyperparameter, and $\widetilde{\boldsymbol{M}}^{(t)}, \widetilde{\boldsymbol{V}}^{(t)}$ is determined by $\boldsymbol{G}^{(t-1)}$, the gradient of $\boldsymbol{W}^{(t-1)}$, and the optimization states $\boldsymbol{M}^{(t-1)}, \boldsymbol{V}^{(t-1)}$. Specifically, we update the $\boldsymbol{M}^{(t-1)}, \boldsymbol{V}^{(t-1)}$ by

$$\boldsymbol{M}^{(t)} = \beta_1 \boldsymbol{M}^{(t-1)} + (1 - \beta_1)\boldsymbol{G}^{(t-1)},$$
$$\boldsymbol{V}^{(t)} = \beta_2 \boldsymbol{V}^{(t-1)} + (1 - \beta_2)\boldsymbol{G}^{(t-1)} \odot \boldsymbol{G}^{(t-1)},$$

where $\beta_1$ and $\beta_2$ are hyperparameters. Then, $\widetilde{\boldsymbol{M}}^{(t)}$ and $\widetilde{\boldsymbol{V}}^{(t)}$ are calculated as

$$\widetilde{\boldsymbol{M}}^{(t)} = \boldsymbol{M}^{(t)}/(1 - \beta_1^t),$$
$$\widetilde{\boldsymbol{V}}^{(t)} = \boldsymbol{V}^{(t)}/(1 - \beta_2^t).$$

### B.2 COSINE ANNEALING LEARNING RATE SCHEDULE

In pretraining, a gradient-based optimization algorithm (Loshchilov, 2017) is employed to update the model. Specifically, at the $t$-th step, we update the matrix $\boldsymbol{W}^{(t-1)}$ from the last step by

$$\boldsymbol{W}^{(t)} = \boldsymbol{W}^{(t-1)} - \gamma^{(t)}\widetilde{\boldsymbol{G}}^{(t)}, \tag{B.1}$$

where $\widetilde{\boldsymbol{G}}^{(t)}$ is the smoothed gradient of matrix $\boldsymbol{W}^{(t-1)}$ (see Appendix B.1), and $\gamma^{(t)}$ is the learning rate, which is often scheduled by a cosine annealing with a linear warm-up (Loshchilov & Hutter,

2016). Specifically, given a fixed total number of training steps $T$, the cosine annealing learning rate schedule is

$$\eta(t; T, \eta_{\mathrm{p}}, \eta_e) = \begin{cases} \frac{t}{T_0}\eta_p & \text{if } 0 < t \leq T_0, \\ c_1 \cos\left(\frac{t-T_0}{T}\pi\right) + c_2 & \text{if } T_0 < t \leq T, \end{cases} \tag{B.2}$$

where $c_1 = \frac{1}{2}(\eta_p - \eta_e), c_2 = \frac{1}{2}(\eta_p + \eta_e), T_0$ represents the number of warmup steps, and $\eta_p$ and $\eta_e$ are the peak and end learning rates, respectively.

## C ITERATIVE PRUNING

### C.1 SENSITIVITY SCORE

For a model with parameters $\boldsymbol{\theta}^{(t)} \in \mathbb{R}^N$, we measure parameter importance by sensitivity (Molchanov et al., 2019):

$$\boldsymbol{\omega}^{(t)} = \left|\nabla\mathcal{L}(\boldsymbol{\theta}^{(t)}) \odot \boldsymbol{\theta}^{(t)}\right|, \tag{C.1}$$

where $\mathcal{L}(\cdot)$ is the loss function. This score represents a the first-order approximation of each parameter's impact on the loss: $\boldsymbol{\omega}^{(t)} \approx \left|\mathcal{L}(\boldsymbol{\theta}^{(t)}) - \mathcal{L}(\boldsymbol{\theta}_{-k}^{(t)})\right|$, where $\boldsymbol{\theta}_{-k}^{(t)} = (\theta_1, \ldots, \theta_{k-1}, 0, \theta_{k+1}, \ldots, \theta_N)$.

### C.2 CUBICAL SPARSITY SCHEDULE IN ITERATIVE PRUNING

The sparsity at the $t$-th step is

$$r(t; T_p) = \begin{cases} 0 & 0 < t \leq T_w, \\ R\left(1 - \frac{t-T_w}{T_p}\right)^3 & T_w < t \leq T_w + T_p, \\ R & T_w + T_p < t \leq T, \end{cases}$$

where $R$ is the target sparsity, and $T_w, T_p, T$ are the number of pruning warmup steps, iterative pruning steps, total iterative pruning steps, respectively. Note that we set $T_w = 0$ if the iterative pruning is in integrated enlarge-and-prune pipelines, and it is set to the same as the learning rate warmup steps if it is in naive enlarge-and-prune pipelines.

## D SUMMARY OF IDEA PRUNE ALGORITHM

We provide a summary of the proposed integrated pruning pipeline as the following algorithm.

---
**Algorithm 1** IDEA Prune
---
    Sparsity schedule $r(t; T_p)$, learning rate schedule $\eta(t; T, \eta_p, \eta_e)$. Randomly initialize model weights $\boldsymbol{W}^{(0)}$. Zero initialize approximate importance scores $\widetilde{\boldsymbol{S}}^{(0)}$. **for** t = 1 to $T$ **do**

4:     Compute gradients $\nabla l_i(\boldsymbol{W}^{(t-1)})$ on a mini batch.
5:     Compute learning rate $\alpha^{(t)} \leftarrow \eta(t; T, \eta_p, \eta_e)$.
6:     Update weights $\boldsymbol{W}^{(t)}$ following (B.1).
7:     Update importance scores $\widetilde{\boldsymbol{S}}^{(t)}$ following (C.1).
8:     Compute neuron importance $\boldsymbol{c}^{(t)}$ following (3.2).
9:     Compute pruning masks $\boldsymbol{m}^{(t)}$ following (3.3).
10:    Prune weights $\boldsymbol{W}^{(t)}$ by masks $\boldsymbol{m}^{(t)}$ following (2.1).
11: **end for** $\boldsymbol{W}^{(T)}$

---

## E MINITRON: ACTIVATION-BASED PRUNING

We denote the activation of the $n$-th data sample of an FFN layer as

$$\boldsymbol{Z}^{(n)} = \sigma(\boldsymbol{X}^{(n)}\boldsymbol{W}_{\mathrm{up}}^{\top}) \odot (\boldsymbol{X}^{(n)}\boldsymbol{W}_{\mathrm{gate}}^{\top}),$$

where $\boldsymbol{Z}^{(n)} \in \mathbb{R}^{l \times h}$, $l$ is the sequence length. The importance score of the $i$-th neuron is defined as

$$\boldsymbol{c}_i = \frac{1}{B} \sum_{n=1}^{B} \|\boldsymbol{Z}_{[:,i]}^{(n)}\|_2,$$

where $B$ is the size of the calibration dataset (a random subset of the pre-training corpus), $\boldsymbol{z}_{[:,i]}^{(n)}$ is the $i$-th column of $\boldsymbol{Z}^{(n)}$, and $||\cdot||_2$ is the L2-norm of a vector. Following Muralidharan et al. (2024), we set $B = 1024$. Finally, we apply (3.3) with $T_p = 1$ to generate the pruning mask.

# F  MODEL ARCHITECTURE AND LEARNING RATE

We present the key configurations of the models in Table 4.

| # Parameter | Input Length | Input Dim | # Attn Heads | Hidden Dim | # Layers | Peak LR | End LR |
|---|---|---|---|---|---|---|---|
| **300M** | 4096 | 1024 | 16 | 3072 | 24 | 7.5e-3 | 3.75e-5 |
| **1B** | 4096 | 2048 | 32 | 6528 | 24 | 3.75e-3 | 1.88e-5 |
| **2.8B** | 4096 | 2048 | 32 | 16384 | 24 | 3.75e-3 | 1.88e-5 |

Table 4:  Model configurations (transposed).

# G  FULL TABLE AND ADDITIONAL RESULTS

We present the full tables that are compressed in Section 5.1 and Section H.2 due to the paper length limit. The full version of Table 2 is Table 5. The full version of Table 8 is Table 6.

| Model | LR Schedule | OWT↓ | Arc-C↑ | Hellaswag↑ | TriviaQA↑ | MMLU↑ |
|---|---|---|---|---|---|---|
| 2.8B-2T@1T | - | 9.99 | 38.7 | 51.6 | 24.8 | 26.0 |
| 1.3B pruned from 2.8B-2T@1T | Resumed | **8.88** | 39.0 | **54.0** | **31.1** | **46.4** |
| | Restarted | 8.94 | 39.7 | 54.5 | 30.1 | 37.0 |
| 2.8B-1T@1T | - | 8.12 | 44.5 | 56.9 | 37.3 | 50.8 |
| 1.3B pruned from 2.8B-1T@1T | Resumed | 8.89 | 39.3 | 53.6 | 31.8 | 38.2 |
| | Restarted | 8.95 | 39.9 | 53.3 | 31.0 | 36.8 |

Table 5: Ablation of the initialization and the learning rate schedule in enlarge-and-prune pipelines. *No Train*: the enlarged model without any further pruning or training. Our integrated enlarge-and-prune pipeline is equivalent to 2.8B-2T@1T initialization with a resumed learning rate schedule. Target size: 1.3B.

## G.1  RESULTS OF FINAL STAGE

We evaluated our model's performance at the final three checkpoints—corresponding to 0.96T, 0.98T, and 1.00T training tokens. The results in Table 7 demonstrate remarkable consistency across these checkpoints. The minimal variations observed typically are within 0.2. This indicates our training enters the convergence regime, and therefore, the noise level remains minimal.

# H  ABALATION AND EXTENTION

## H.1  ROBUSTNESS OF HYPERPARAMETERS

We also study how the pruning schedule, importance score combination functions, and the moving average coefficient affect IDEA Prune. The following experiments show these factors are robust, which does not cost much tuning effort.

| Method | | OWT↓ | Arc-C↑ | Hellaswag↑ | TriviaQA↑ | MMLU↑ |
|---|---|---|---|---|---|---|
| OTRP | Naive | 8.980 | 38.9 | 53.4 | 29.6 | 32.5 |
| | Unified | 8.926 | 41.1 | 53.8 | 30.1 | 42.9 |
| Minitron | Naive | 8.973 | 38.7 | 53.6 | 30.7 | 31.4 |
| | Unified | 8.916 | 39.8 | 53.6 | 30.8 | 43.9 |
| Sheared LLaMA | Naive | 8.960 | 39.4 | 53.6 | 30.1 | 33.4 |
| | Unified | 8.892 | 43.2 | 54.0 | 30.5 | 42.4 |

Table 6: Extension of the the integrated enlarge-and-prune pipeline with OSPR, Minitron, and Sheared LLaMA pruning methods. We prune a 2.8B model to 1.3B with 2T tokens. We report the perplexity on OpenWebText, the average accuracy on comprehensive tasks, and 5-shot accuracy on MMLU.

| Eval at (tokens) | OWT ↓ | ARC-C ↑ | HellaSwag ↑ | TriviaQA ↑ | MMLU↑ |
|---|---|---|---|---|---|
| 0.96T | 9.10 | 39.59 | 52.84 | 30.03 | 28.72 |
| 0.98T | 9.10 | 38.40 | 52.69 | 30.56 | 29.12 |
| 1.00T | 9.10 | 39.76 | 52.81 | 30.33 | 28.95 |
| Average | 9.10 | 39.25 | 52.78 | 30.31 | 28.93 |

Table 7: Performance across the last three checkpoints during training.

**Pruning Schedule.** We investigate the impact of pipeline schedule on model performance, using a 400B token budget. We vary pretraining step proportions at 10%, 30%, and 50%, and examine different pruning steps $T_p$ used in $r(t; T_p)$. Figure 3a reveals robust performance across a wide range of steps. For instance, with 50% steps in pretraining, valid pruning steps extend from 10% to 70%. The analysis suggests an optimal pruning proportion around 70% for a fixed pretraining stage, which differs from the main experiments in Section 4.2, indicating potential for further optimization in IDEA Prune.

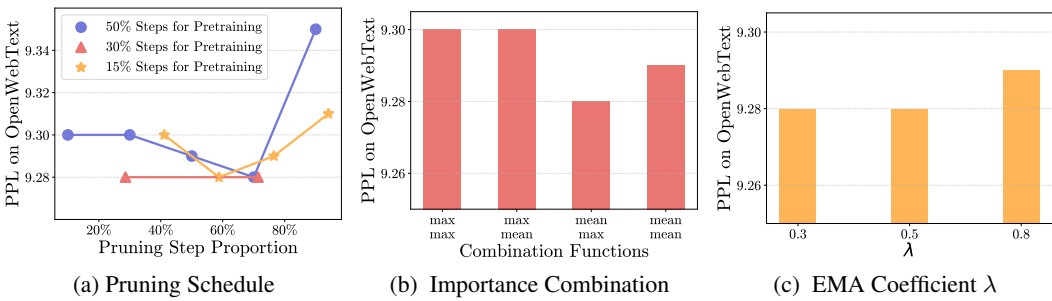

(a) Pruning Schedule   (b) Importance Combination   (c) EMA Coefficient $\lambda$

Figure 3: Hyperparameter study. Figure 3a shows the a wide range of pipeline schedule works for the best performance. Figure 3b implies "mean-max" is the best choice of $f_1$ and $f_2$, though the others do not show significant degradation. Figure 3c shows the exponential moving average coefficient $\lambda$ is robust in a wide range from 0.3 to 0.8.

**Importance Score Combination.** We study how the choice of importance score combination function $f_1(\cdot)$ and $f_2(\cdot)$ in (3.2) affects the model performance. We try $\max(\cdot)$ and $\mathrm{mean}(\cdot)$ for each of $f_1(\cdot)$ and $f_2(\cdot)$. As shown in Figure 3b, "mean-max" is the best choice of $f_1$ and $f_2$, though the others do not show significant degradation.

**Moving Average Coefficient.** We investigate the impact of the coefficient $\lambda$ in (3.1), exploring values of 0.3, 0.5, and 0.8. A lower coefficient tends to favor current-step importance scores. Figure 3c indicates the optimal $\lambda$ lies between 0.3 and 0.5, with 0.8 also acceptable.

## H.2 EXTENSION OF INTEGRATED PIPELINE

We demonstrate that our integrated pipelines are extensible beyond the proposed iterative structured pruning method. For OSRP and Minitron, we apply one-shot pruning to the intermediate checkpoint (2.8B-2T@1T). For Sheared LLaMA, we perform mask learning initialized from the same checkpoint using two distinct learning rate schedules: a resumed schedule for weight matrices and a fully decaying schedule for auxiliary parameters. We conduct recovery training for the pruned models using the resumed learning rate schedule described in Section 5.1.

As shown in Table 8, the integrated enlarge-and-prune pipeline shows improvement over the naive enlarge-and-prune pipeline across all pruning methods. The improvement on MMLU is the most significant, which implies the rising learning rate in naive enlarge-and-prune pipelines causes the knowledge loss the most.

| Method | Pipeline | OpenWebText↓ | Comp Avg ↑ | MMLU ↑ |
|---|---|---|---|---|
| OSRP | Naive | 8.98 | 40.6 | 32.5 |
| | Integrated | **8.93** | **41.7** | **42.9** |
| Minitron | Naive | 8.97 | 41.0 | 31.4 |
| | Integrated | **8.92** | **41.4** | **43.9** |
| Sheared LLaMA | Naive | 8.96 | 41.0 | 33.4 |
| | Integrated | **8.89** | **42.6** | **42.4** |

Table 8: Extension of the the integrated enlarge-and-prune pipeline with OSPR, Minitron, and Sheared LLaMA pruning methods. We prune a 2.8B model to 1.3B with 2T tokens. We report the perplexity on OpenWebText, the average accuracy on comprehensive tasks, and 5-shot accuracy on MMLU. The best results are in **bold**.

# I TOKEN BUDGET OR FLOPS BUDGET

Pruning pipelines intrinsically consume more FLOPs than training the target-sized small model given the same training tokens. Therefore, we compare the pruning pipelines with the training from scratch baselines under the same FLOPs as well.

The results in Figure 4 show our proposed pruning pipeline outperforms the naive pipeline at all FLOPs budgets, and the training from scratch baseline at higher FLOPs budgets, which is practical under our interested scenario when practitioners have sufficient computational resources.

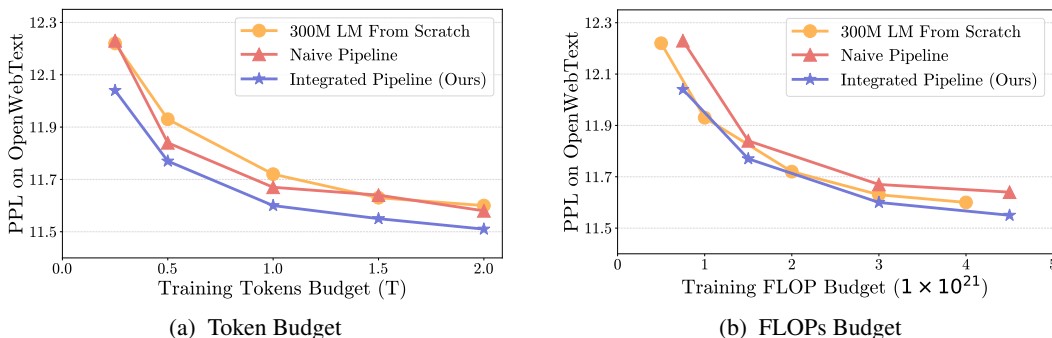

(a) Token Budget          (b) FLOPs Budget

Figure 4: **Left**: Perplexity on OpenWebText of 300M model trained with multiple training token budgets from 0.25T to 2T. **Right**: We calculate the FLOPs of each point from the left figure and convert the x-axis to FLOPs.

