# OpenReview forum: "On the Efficiency of Structured Pruning in Small Language Model Pretraining"
_ICLR.cc/2026/Conference — Submitted to ICLR 2026_

### Official Review · Reviewer_aXpd · 2025-10-22

**Soundness:** 3
**Presentation:** 2
**Contribution:** 2
**Rating:** 6
**Confidence:** 3

**Summary:**

This paper proposes IDEA Prune, an integrated enlarge-and-prune pipeline that unifies model training, pruning, and recovery under a single learning rate schedule. The method shows improvements over naive pipelines and reveals important training dynamics

**Strengths:**

1. The paper is well written and targets the practical case of "abundant training resources but strict deployment constraints" and makes a convincing case that training dynamics impact pruning results beyond just the pruning method itself.
2. Well-Designed Ablations with Interesting Findings： Section 5 provides a thorough examination of initialization, learning rate schedules, and model size factors. The paper identifies 2.6× FFN width as optimal, demonstrating robustness across a wide 1.3×-5.3× range and demonstrating an interesting synergy where pruning combined with knowledge distillation outperforms direct KD training from scratch.

**Weaknesses:**

1. All main experiments use only 2.8B compression, which cannot validate whether findings generalize to larger models. Does IDEA Prune work for 7B/3B compression? Would the 2.6× optimal enlarged size also hold for larger models, or does it scale with model size? Do intermediate checkpoints remain superior at larger scales?
2. Missing Ablations： No ablation study is provided to compare different token allocation strategies and iterative pruning step frequency.
3. The paper overlooks several critical factors established in prior work: **Learning Rate**: Prior research [1] shows that peak and end learning rates significantly influence pruning outcomes, yet this paper does not examine their effects. **Optimizer**: All experiments use only Adam without testing whether findings generalize to other optimizers. **Learning Rate Schedule**: The paper only compares cosine annealing while overlooking Warmup-Stable-Decay (WSD) [2], which outperforms cosine for small LLM training (and board using for current LLMs). Including WSD would strengthen the paper's claims.

[1] Sparse Training via Boosting Pruning Plasticity with Neuroregeneration

[2] MiniCPM: Unveiling the Potential of Small Language Models with Scalable Training Strategies

**Questions:**

The key questions raised in the Weaknesses section above

---

> ### Author Response · Authors · 2025-11-20
> **Response 1**
>
> Dear review, we appreciate your support and reviews. We address your concerns as follows.
>
> ### **Weakness 1**
> > Does IDEA Prune work for 7B/3B compression?
>
> Our scenario focuses on how to obtain best possible on-device models, which is usually less than 3B, and therefore, scaling up beyond 3B is out of our scope. Please refer to the Common Comments to see why 3B is a reasonable size for on-device models.
>
> Even though scaling beyond 3B would bring more insights and interesting observations, it is too expensive to conduct experiments to comprehensively compare pruning pipelines and training from scratch baselines. If experiments of 7B/3B with 2T tokens were conducted, an estimate cost could be 38 days and $1M with 16*8 H100 GPUs. Moreover, 2T or even more tokens are necessary because we are under the scenario where training small models with longer tokens becomes less and less token efficient (the scaling law), and want to know if enlarge-and-prune pipelines are still token efficient under the extreme scenario.
>
> > Would the 2.6× optimal enlarged size also hold for larger models, or does it scale with model size?
>
> We would not know if the optimal enlargement factor scales with model size, as we only had the sweep on 300M models. However, in Section 5.2 (Line 426 to Line 428), we have shown IDEA Prune with a wide range of enlargement factors outperforms training a small model from scratch. This shows whether or not 2.6x is an optimal factor for larger models (e.g., 2.8B to 1.2B settings in our experiment) does not affect the conclusion that IDEA Prune outperforms training small models from scratch. It only matters how better it can be.
>
> > Do intermediate checkpoints remain superior at larger scales?
>
> Could you please clarify “intermediate checkpoints remain superior”? We don’t think we made this conclusion. Instead, we find we don’t need the intermediate checkpoint to be superior. The intermediate checkpoints has been trained only with a small portion of tokens (30% of the total token budget). We have discussed this in Section 5.1 (Line 407).
>
> ### **Weakness 2**
> Due to the page limit, we defer the ablation study of token allocation and pruning frequency in Appendix H.1, and make reference at the beginning of Section 5 (Line 367).
>
> In Figure 3(a) , we find 30% tokens for enlarged model pretraining and 70% for pruning + recovery is the optimal allocation. In the pruning + recovery stage, 60%-80% tokens for pruning is the optimal allocation, which means the pruning step frequency should be slow.
>
> We will either move the ablation study to the main paper or make a more detailed reference at the beginning of Section 5 in the next edition.
>
> ### **Weakness 3**
> **Learning rate**
>
> We agree that continuing to carefully tune the peak and end learning rates for pruning can potentially further improve the performance, but it doesn’t affect the conclusion that IDEA Prune works better than baselines that have been well tuned. We have discussed this in Section 4.2 (Line 352-360)
>
> **Optimizer and Learning Rate Schedule**
>
> We agree that a broader coverage of optimizers and learning rate schedulers would further strengthen the study. However, our experiments are already extremely costly to run: pretraining a 2.8B/1.2B models for 2T tokens. A full grid over multiple optimizers × schedulers × pruning pipelines would require substantial additional GPU budget beyond our current resources.
>
> Given this constraint, we intentionally adopt the mainstream configuration used in many current LLMs: Adam/AdamW with warmup plus cosine decay. For example, Llama, Qwen, and MobileLLM all report using Adam-family optimizers with cosine-style schedules for pretraining. We believe this makes our results directly relevant to common practice.

---

> > ### Comment · Reviewer_aXpd · 2025-11-21
> >
> > Dear Authors,
> >
> > Thank you for your detailed reply and for explaining the motivations and scenarios. I now understand your choice of model size.
> > I still have a bit of confusion. In your examples (Samsung and Google), the on-device models are distilled from large models the companies already have. So in most real cases, companies build device models from existing large models, not by pretraining a big model that they never plan to deploy. I’m not sure whether we really need to consider the pretraining cost in this way. Would this assumption still make sense?
> > Also,  many open-source large models are free for commercial use,  isn’t the teacher basically “free”?
> >
> >
> > Please correct me if I misunderstood anything.

---

> > > ### Author Response · Authors · 2025-11-24
> > > **Response 2**
> > >
> > > Dear reviewer, we appreciate your quick feedback. We understand you have confusions about the "existing large models" in our assumption. We'd like to justify our assumption by breaking down the "existing large models" into internal v.s. open-source existing large models and medium (around 7B) v.s. very large (>20B) existing large models.
> > >
> > > > “Would this assumption still make sense?”
> > >
> > > In the Samsung example, they only deploy the on-device model, so the existing large models, source models for pruning or teacher models for distillation, are typically not deployed on device. Therefore, their cost should be included for shipping the on-device model.
> > >
> > > In the Google example, the teacher model may be deployed, but it is unlikely to be the source model for pruning because it is usually too large to be a good direct pruning source. For example, the cloud-deployed models are often greater than 20B, but the on-device model is less than 3B. Pruning 20B to 3B often incurs severe performance degradation, so this is typically not a realistic pruning pipeline. In this case, a realistic choice is to either train a 3B model from scratch, or apply the pruning pipeline by training a 7B model and pruning it to 3B (both cases can use distillation with the deployed 20B teacher model). That 7B model is not free; it must be trained, and that cost is exactly what our paper is analyzing.
> > >
> > > **In summary**, existing internal models with medium size (about 7B) are often not deployed on device and are often for pruning purposes, so the cost should be counted. Existing internal models with large size (>20B) are typically too large to serve as direct pruning sources for ≲3B targets, so we still need a medium-size model for pruning and the training cost of the medium-size model should be included.
> > >
> > >
> > > > “Also, many open-source large models are free for commercial use, isn’t the teacher basically “free”?”
> > >
> > > Very large open-source models (e.g., >20B) can be suitable as teachers for distillation, but not as pruning sources, due to the large size gap to 3B. So even with “free” large teachers, we still need to decide whether to pretrain a 7B source model for pruning or train a 3B model from scratch.
> > >
> > > However, open-source models are often data- and domain-misaligned. Medium-size open-source models (≈7B) are trained on broad general-domain data, while many on-device use cases are domain- or product-specific. Given the limited capacity of 3B-scale models, continuing to train a general 7B model (or its pruned 3B variant) can be less effective than training 3B models from scratch, because the model capacity is already “used up” by generic content that may not be relevant.
> > >
> > > Moreover, companies seek full control over training data and stack. For example, even though Qwen families are free for commercial use, Google still builds the Gemini-nano based on their own Gemini models instead of the Qwen families. And this is also true for Gemma and Llama families.

---

> > > > ### Comment · Reviewer_aXpd · 2025-11-26
> > > >
> > > > Thank you for the further clarification. Personally, I like the idea of this work. My main concern is the experimental setting. Currently, the results are comparable to target-size models trained from scratch, which feels a bit limited. I get that there are computational constraints, but I think more comprehensive experiments would make the claims more convincing. That said, I remain positive about this work and keep my current score.

---

### Official Review · Reviewer_an3P · 2025-10-26

**Soundness:** 2
**Presentation:** 2
**Contribution:** 2
**Rating:** 4
**Confidence:** 4

**Summary:**

The paper explores how to most effectively train small generative language models (SLMs) when training resources are abundant but inference constraints, such as memory and latency, are strict. To address these issues, the paper introduces IDEA Prune (IntegrateD Enlarge-And-Prune), a unified training framework that integrates model enlargement, pruning, and recovery into a single pretraining schedule.

**Strengths:**

1. The paper introduces IDEA Prune (IntegrateD Enlarge-And-Prune), a new, unified training pipeline that merges model enlargement, pruning, and recovery within a single cosine annealing schedule. This integration avoids the discontinuities and knowledge forgetting observed in conventional two-stage pipelines. Moreover, the method smoothly transitions between pretraining and pruning without resetting optimization states.

2. The analysis section provides several counterintuitive yet insightful findings, such as: intermediate checkpoints, though weaker in absolute performance, yield better pruning outcomes than final checkpoints due to better learning rate alignment. Moreover, the learning rate schedule affects performance more than initialization quality.

**Weaknesses:**

1. The core components of IDEA Prune, iterative structured pruning, importance scoring via sensitivity, and cosine-annealed learning rate schedule are well-established in prior literature [1]. The paper’s main contribution is integrating existing techniques into a unified pipeline rather than proposing fundamentally new algorithms or pruning criteria. All technical differences are only a recombination of known ideas under a specific training regime.

2. The study exclusively focuses on pruning FFN width and excludes attention layers and hidden size, which are all included in Minitron and ShearedLLaMA paper.

3. How to decide the optimal hyperparameter configuration in the main experiments? The authors explicitly state that the token budget allocation (50% for enlarged pretraining, 50% for pruning + recovery) and learning rate settings in the main experiments were not tuned for peak performance but chosen to enable fair comparison with naive baselines.

[1] Sreenivas S T, Muralidharan S, Joshi R, et al. Llm pruning and distillation in practice: The minitron approach[J]. arXiv preprint arXiv:2408.11796, 2024.

**Questions:**

See Weakness

---

> ### Author Response · Authors · 2025-11-19
> **Response 1**
>
> Dear reviewer, thank you for your time and reviews. We address your concerns as follows.
>
> ### **Weakness 1**
> Our contribution goes beyond combining known components. First, we pose and study a new practical question: when training resources are sufficient but inference budgets are strict, is it worth pretraining a larger model that is never deployed once we include its pretraining cost? This problem is not studied in [1].
>
> Second, to answer the question, we propose a new framework: the IDEA Prune pipeline, where the main technical novelty lies in the integrated learning rate schedule to bridge the gap between the pretraining and pruning stages. This bridge is not studied in [1] as they don’t even have the two-stage training framework.
>
> As to the pruning technique you are concerned about, our IDEA Prune is adaptable to multiple pruning techniques (activation-based, mask-learning, and random pruning), underscoring its general applicability, as demonstrated in Appendix H.2. Moreover, while iterative pruning techniques have indeed been explored previously, we tailor the iterative pruning specifically to complement this integrated pipeline by Eq 3.2 and conduct ablation experiments in Figure 3(b) in Appendix H.1 to contribute a best practice to the community.
>
> ### **Weakness 2**
>
> In the enlarge-and-prune pipeline (including the separate pipeline and our IDEA Prune), what part to prune is decided by what part we want to enlarge. In our current design, we only enlarge the FFN. Therefore, we are only able to prune FFN.
>
> There are three reasons why we don't enlarge the MHA or the layers.
>
> First, [2] suggest pruning width (i.e., the hidden dimension of FFN and the number of attention heads) is more efficient than pruning depth (i.e., the number of transformer layers). As a result, we should enlarge the width (the hidden dimension of FFN and the number of attention heads) instead of the depth. We have discussed this in Section 2.2 (Line 177).
>
> Second, the attention layers only have minor contributions to the total parameters (less than 20%). Therefore, enlarging MHA does not enlarge the model size by a lot compared to FFN. For example, enlarging the MHA by 2x only enlarges the entire model by 20.2 + 0.8 = 1.2x, but enlarging the FFN by 2x enlarges the entire model by 0.2 + 20.8 = 1.8x. We have discussed this in Section 2.2 (Line 180).
>
> Last, we start with this simple design, and this simple design has already been effective: IDEA Prune with only enlarging FFN has already outperformed the small model trained from scratch. We admit that it is possible that enlarging MHA or layers may bring additional improvement, but exhaustively finding an optimal solution is too expensive and is not the priority of this paper, so we leave this study as future work. We will add a future work section in the next edition.
>
> ### **Weakness 3**
>
> We conduct ablation study in Section H.1, where we identify 30% for enlarged pretraining and 70% for pruning + recovery as the optimal allocation, but for the main experiments we stick to 50% + 50% for fair comparisons to baselines. Even though the token budget allocation and the learning rate are suboptimal in IDEA Prune, it still outperforms baselines that have been well tuned. We agree that continuing to carefully tune the peak and end learning rates for pruning can potentially further improve the performance, but exhaustively finding an optimal configuration is too expensive and is not the priority of this paper, so we leave this study as future work. We have discussed this in Section 4.2 (Line 352-360).
>
> [2] Saurav Muralidharan, et al. Compact language models via pruning and knowledge distillation.

---

> ### Author Response · Authors · 2025-11-25
>
> Dear reviewer, we'd like to kindly follow up on our responses to your reviews. We would greatly appreciate it if you can let us know whether our responses address your concerns or if any further clarification is needed.

---

### Official Review · Reviewer_Q9bX · 2025-10-31

**Soundness:** 2
**Presentation:** 2
**Contribution:** 2
**Rating:** 4
**Confidence:** 4

**Summary:**

This paper studies the pretraining of small LMs using structured pruning. The authors argue that, when accounting for the pretraining cost of the large source model, the conventional pipeline (which prunes an existing large model and retrains it for performance recovery) becomes inefficient. To address this, they propose an enlarge-and-prune pipeline and compare it with both conventional pruning pipelines and training small LMs from scratch under an equivalent training token budget. Their analysis shows that the integrated approach, which adopts iterative structured pruning and a single cosine learning rate schedule, can produce more effective small LMs.

**Strengths:**

- This work revisits existing pruning–retraining pipelines and provides a controlled analysis comparing multiple scenarios for obtaining small LLMs.
- The proposed method, including its learning rate scheduling and FFN neuron pruning strategy, is simple and easily adaptable.
- The paper is clearly written and easy to follow.

**Weaknesses:**

- While the paper's flow and presentation are clear, the main motivation "Is it worth pretraining a large model even if it is never deployed?" seems somewhat unrealistic. In practice, a common way to obtain small LMs is by pruning or distilling already existing large models, rather than by pretraining a new large model solely for the purpose of pruning. The experimental setup does not include comparisons with existing pretrained large models (e.g., LLaMA-3, Qwen-3) pruned to comparable small model sizes to those studied in this work under an equivalent training budget, which would better reflect practical real-world settings.
- The study focuses solely on FFN pruning, possibly due to the ease of applying it in an enlarge-prune setting. Nevertheless, prior research (e.g., https://arxiv.org/abs/2405.18218) has shown that attention layers can often be more prunable than FFN layers, and many recent LLM pruning approaches jointly prune MHA and FFN, or even remove entire layers or transformer blocks. Could the authors also consider MHA and/or layer pruning, which are widely adopted structured pruning strategies?
- While the proposed method shows improvements over the conventional pruning-retraining pipelines in Table 1, the performance gains appear relatively modest and not clearly substantial compared to strong baselines such as Sheared LLaMA and Minitron. Moreover, Table 1 reports results only for a single target scale (1.3B), which limits the generality of the findings. It would be valuable to examine whether the observed trends hold for other small model sizes (e.g., 0.7B, 2B). In addition, the study considers only a 2.8B enlarged source model; it remains unclear how the method would behave when pruning from larger source models, which could provide further insights into scaling behavior.
- The analysis on the impact of training token budget is informative. However, since the proposed method focuses on structured pruning, it would also be valuable to provide an analysis of the practical inference-time speedup and memory footprint reduction. Such results would strengthen the empirical evaluation by demonstrating the real-world efficiency benefits.
- Although the paper claims to adopt iterative pruning, the main text does not clearly illustrate the prune-retrain iteration mechanism. If Algorithm 1 (currently in Appendix D) were included in the main paper, readers could more easily understand that pruning and retraining are alternated within each training step, making the 'iterative' nature of the method explicit.

**Questions:**

Please refer to the weakness section.

---

> ### Author Response · Authors · 2025-11-19
> **Response 1**
>
> Dear reviewer, thank you for your time and review. We address your concerns as follows.
>
> ### **Weaknesses 1**
>
> **About motivation**
>
> We agree that a common way to obtain small LMs is by pruning or distilling existing open-source large LMs. That scenario exists, and it’s valuable.
>
> However, we are studying a totally different problem under a totally different scenario that has already clearly existed: tech companies have sufficient training compute but strict inference budgets; and, due to the license restriction, they are not able to build upon open-source LLMs. Therefore, they have to obtain an on-device model from scratch.
>
> Moreover, training large but serving small is common among tech companies. For example, Samsung describes that their on-device models are pruned and distilled from a “proprietary” internal model developed by Samsung Research. Google’s Gemini Nano is distilled from larger Gemini models.
>
> In the Samsung example, they only deploy the on-device model (no cloud-deployed large models are mentioned in their report), so the internal large models, source models for pruning or teacher models for distillation, are typically not deployed on device. Therefore, their cost should be included for shipping the on-device model. And this is the main scenario we are working on.
>
> In the Google example, the teacher model may be deployed as an API, but it is unlikely to be the source model for pruning because it is usually too large to be a good direct pruning source. For example, the cloud-deployed models are often greater than 20B, but the on-device model is less than 3B. Pruning 20B to 3B often incurs severe performance degradation, so this is typically not a realistic pruning pipeline. In this case, a realistic choice is to either train a 3B model from scratch, or apply the pruning pipeline by training a 7B model and pruning it to 3B (both cases can use distillation with the deployed 20B teacher model). That 7B model is not free; it must be trained, and that cost is exactly what our paper is analyzing.
>
> In summary, existing internal models with medium size (about 7B) are often not deployed on device and are often for pruning purposes, so the cost should be counted. Existing internal models with large size (>20B) are typically too large to serve as direct pruning sources for ≲3B targets, so we still need a medium-size model for pruning and the training cost of the medium-size model should be included.
>
>
> **About experiment set ups**
>
> Since we are studying the problem of how to obtain the best possible on-device model from scratch, pruning existing models like LLaMA-3 or Qwen3 is out of the scope.
>
> There is another reason why we compare IDEA prune to separate enlarge-and-prune pipelines with baseline pruning techniques from scratch instead of LLaMA-3 or Qwen3: we aim to eliminate the factor of the pre-training corpus and the difference of the number of training tokens to provide a clean and fair comparison that only differs in pruning pipelines. Both Qwen3 and Llama have been trained with more than 20T tokens with close-source pretraining corpus. It would not be clear what the most important factor in pruning pipeline would be if we pruned Qwen3 and Llama models.
>
> ### **Weaknesses 2**
>
> In the enlarge-and-prune pipeline (including the separate pipeline and our IDEA Prune), what part to prune is decided by what part we want to enlarge. In our current design, we only enlarge the FFN. Therefore, we are only able to prune FFN.
>
> There are three reasons why we don't enlarge the MHA or the layers.
>
> First, [1] suggest pruning width (i.e., the hidden dimension of FFN and the number of attention heads) is more efficient than pruning depth (i.e., the number of transformer layers). As a result, we should enlarge the width (the hidden dimension of FFN and the number of attention heads) instead of the depth. We have discussed this in Section 2.2 (Line 177).
>
> Second, the attention layers only have minor contributions to the total parameters (less than 20%). Therefore, enlarging MHA does not enlarge the model size by a lot compared to FFN. For example, enlarging the MHA by 2x only enlarges the entire model by 2*0.2 + 0.8 = 1.2x, but enlarging the FFN by 2x enlarges the entire model by 0.2 + 2*0.8 = 1.8x. We have discussed this in Section 2.2 (Line 180).
>
> Last, we start with this simple design, and this simple design has already been effective: IDEA Prune with only enlarging FFN has already outperformed the small model trained from scratch. We admit that it is possible that enlarging MHA or layers may bring additional improvement, but exhaustively finding an optimal solution is too expensive and is not the priority of this paper, so we leave this study as future work. We will add a future work section in the next edition.
>
> [1] Saurav Muralidharan, et al. Compact language models via pruning and knowledge distillation.

---

> ### Author Response · Authors · 2025-11-19
> **Response 2**
>
> ### **Weakness 3**
>
> > While the proposed method shows improvements over the conventional pruning-retraining pipelines in Table 1, the performance gains appear relatively modest and not clearly substantial compared to strong baselines such as Sheared LLaMA and Minitron.
>
> First, in Table 1, we should compare all pruning pipelines (OSPR, Minitron, ShearedLlama, IDEA Prune) with training a small model from scratch (1.3B-2T from scratch) to answer the question we pose: is it worth pretraining a large model even if it is never deployed? In this comparison, Minitron and ShearedLlama underperform the 1.3B-2T from scratch, but IDEA Prune outperforms the 1.3B-2T from scratch. This indicates the answer to the question depends on pruning pipelines. We have discussed this in Section 4.1 (Line 347-351).
>
> Second, while the performance gains of IDEA Prune over small models from scratch may appear modest, they are meaningful in the context of small language model pretraining, which tend to saturate quickly due to their limited capacity when given sufficiently large training tokens (2T in our setting). We have discussed this in Section 7 (Line 478-485).
>
> > Moreover, Table 1 reports results only for a single target scale ... could provide further insights into scaling behavior.”
>
> The generality of our findings is not limited. Within 3B, we have taken experiments on both 600M to 300M and 2.8B to 1.3B settings, and shown our enlarge-and-prune pipeline works better than 1) training a small model from scratch and 2) a naive separate pruning pipeline given the same and sufficiently large training tokens. The experiment of 600M to 300M is in Figure 1(a), and the experiment of 2.8B to 1.3B is in Table 1.
>
> In addition, our scenario focuses on how to obtain best possible on-device models, which is usually less than 3B (see why 3B is a reasonable size in Common Comments), and therefore, **scaling up beyond 3B is out of our scope**. Even though scaling beyond 3B would bring more insights and interesting observations, it is too expensive to conduct experiments to comprehensively compare pruning pipelines and training from scratch baselines. If one training of 7B/3B with 2T tokens were conducted, an estimate cost would be 38 days and $1M with 16*8 H100 GPUs.
>
> ### **Weakness 4**
> The problem we study is NOT how to compress an existing large (2.8B) model into a small (1.3B) model to speed up inference or reduce memory footprint. Rather, it is how to obtain a small (1.3B) model from scratch.  IDEA Prune enlarges the small (1.3B) model to larger size (2.8B) and prunes it back to small size (1.3B). Therefore, our paper does not claim any inference speedup or memory footprint reduction. Besides, since the pruned model has the identical model architecture as the small model, they have the same inference speed.
>
> ### **Weakness 5**
> Thank you for the suggestion. Due to the page limit, we are not able to put the algorithm in the main paper, but we will move it to the main paper or provide a brief description of the iterative nature in the next edition.

---

> ### Author Response · Authors · 2025-11-25
>
> Dear reviewer, we'd like to kindly follow up on our responses to your reviews. We would greatly appreciate it if you can let us know whether our responses address your concerns or if any further clarification is needed.

---

> ### Comment · Reviewer_Q9bX · 2025-11-26
>
> Thank you for the detailed rebuttal. As many reviewers (including myself) have questioned the motivation (i.e., the necessity of considering the pre-training cost of the source model), the authors provide clarifications with examples such as Samsung and Google. However, I still remain unconvinced. It is unclear to me whether these companies trained large models solely to distill or prune them into smaller ones; large models can also be used for many other purposes, such as API-based applications. Therefore, I feel that the current justification for the motivation is not fully convincing.
>
> Although the authors describe certain points as out-of-scope, I believe some of these questions could still have been addressed empirically. For example, if we prune existing publicly available models like LLaMA-3 or Qwen-3 to match the architecture used in this paper, and then fine-tune them on the same data used in this work, would the performance be better or worse? If the performance improves, then there is little reason not to use a pretrained backbone.
>
> Similarly, the authors mention FFN-only pruning and model-size scalability, but providing experimental evidence would have strengthened the paper.
>
> I will therefore maintain my current score.

---

> ### Author Response · Authors · 2025-11-27
>
> Dear reviewer, thank you for the feedback. We would like to first clarify why we assume that we don't have the access to any open-source models. We hope this can help to clarify our motivation and the real scenario we are facing. Second, we'd like to answer your question about how the tech companies really do. Last, we'd like to address your concerns about the FFN-only and model-size scalability.
>
> **Clarifying our scenario and why open-source models are excluded.**
>
> Our paper is written from the perspective of a company that ships and maintains its own on-device models. In this setting, **it is common not to build upon open-source weights** (e.g., LLaMA-3, Qwen-3), but instead to train proprietary models from scratch to retain full control over training data, safety filters, and long-term product roadmaps. Major companies such as Meta (Llama), Google (Gemma/Gemini), NVIDIA (Nemotron), Amazon (Olympus/Nova) and Samsung (Gauss) have all invested heavily in their own foundation models rather than relying on public open-weight models. This demonstrates that “train your own stack” is not hypothetical but already the dominant pattern for large industrial actors. Under this perspective, assuming that open-source backbones are not available or acceptable for a given product line is realistic.
>
> > “It is unclear to me whether these companies trained large models solely to distill or prune them into smaller ones; large models can also be used for many other purposes, such as API-based applications.”
>
> In the Samsung example. They only deploy the on-device model (**no cloud-deployed large models** are mentioned in their report), so the internal large models, source models for pruning or teacher models for distillation, are not deployed on device. Therefore, their cost should be included for shipping the on-device model. **And this is the exact scenario that our paper studies.**
>
> In the Google example, the teacher model may be deployed as an API, but it is unlikely to be the source model for pruning because it is usually **too large to be a good direct pruning source**. For example, the cloud-deployed models are often greater than 20B, but the on-device model is less than 3B. Pruning 20B to 3B often incurs severe performance degradation, so this is typically not a realistic pruning pipeline. In this case, a realistic choice is to either train a 3B model from scratch, or apply the pruning pipeline by training a 7B model and pruning it to 3B (both cases can use distillation with the deployed 20B teacher model). That 7B model is not free; it must be trained, and that cost is exactly what our paper is analyzing. **This scenario is not exactly what we are interested in, but can be an example to justify why we still need a medium source model for pruning even if we have bigger cloud-deployed models.**
>
> **In summary**, existing internal models with medium size (about 7B) are often not deployed on device and are often for pruning purposes, so the cost should be counted. Existing internal models with large size (>20B) are typically too large to serve as direct pruning sources for ≲3B targets, so we still need a medium-size model for pruning and the training cost of the medium-size model should be included.
>
> > "If we prune existing publicly available models like LLaMA-3 or Qwen-3 ... then there is little reason not to use a pretrained backbone."
>
> We do not consider any open-source models, and the main reason is that tech companies want the full control of the training data. We have discussed this at the first beginning of this reply. Therefore, the comparison you suggest does not answer the problem we pose at the beginning of our paper: what is the most effective approach to obtain the best possible small generative language model from scratch?
>
> > “Similarly, the authors mention FFN-only pruning and model-size scalability, but providing experimental evidence would have strengthened the paper.”
>
> We have discussed why we choose to only enlarge FFN in the last response. Adding experiments of enlarging MHA or depth **only extend** our claim; it does **not strengthen** the claim we have already made in this paper.
>
> Regarding “model-size scalability”, **our goal is not to establish a scaling law**. In LLM work, “scaling” typically means moving across orders of magnitude (e.g., 7B→70B→200B). By contrast, adding 0.7B or 2B would be interpolation inside one regime rather than a scaling-law study. Therefore, adding these points does not bring any fundamental insight to the problem we study but only brings additional cost.

---

### Official Review · Reviewer_U8L8 · 2025-11-01

**Soundness:** 3
**Presentation:** 3
**Contribution:** 2
**Rating:** 2
**Confidence:** 3

**Summary:**

This paper investigates whether structured pruning remains efficient when accounting for the full training cost of LLMs. It finds that traditional enlarge-and-prune pipelines lose token efficiency once the large model’s pretraining cost is included, motivating a new IDEA Prune method that integrates enlargement, pruning, and recovery under a single learning-rate schedule. Experiments show that IDEA Prune achieves smoother knowledge retention and superior performance compared to naive pipelines or training small models from scratch, offering practical guidance on building compact yet powerful language models.

**Strengths:**

1. The paper presents an inspiring idea by firstly considering token efficiency during the pre-training stage of LLMs, offering a new perspective on the practicality of structured pruning.

2. It proposes a novel enlarge-and-prune pipeline that effectively eliminates the loss gap during training. The method is simple yet effective.

**Weaknesses:**

1. The underlying assumption of this paper seems somewhat unrealistic. In most practical scenarios, users do not pre-train an LLM from scratch that is never deployed; instead, they typically rely on open-source LLMs. As a result, users neither consider the training cost nor have access to the learning rate. Therefore, I have reservations about the paper’s practical applicability.


2. The experimental validation is also not sufficiently solid, as it only evaluates a single setting where a 2.8B model is compressed to 1.3B. Hence, I remain skeptical about whether the proposed method would remain effective for larger LLMs or under higher compression ratios.

**Questions:**

1. Do we need to consider the pre-training cost?

2. Does the proposed method remain effective for larger LLMs or under higher compression ratios?

---

> ### Author Response · Authors · 2025-11-19
> **Response 1**
>
> Dear reviewer, thank you for your time and review. We address your concerns as follows.
>
> ### **Weakness 1 & Question 1**
>
> > The underlying assumption of this paper seems somewhat unrealistic ... As a result, users neither consider the training cost nor have access to the learning rate
>
> We agree that a common way to obtain small LMs is by pruning or distilling existing open-source large LMs. That scenario exists, and it’s valuable.
>
> **However, we are studying a totally different problem under a totally different scenario that has already clearly existed**: tech companies have sufficient training compute but strict inference budgets; and, due to the license restriction, they are not able to build upon open-source LLMs. Therefore, they have to obtain an on-device model from scratch.
>
> Moreover, training large but serving small is common among tech companies. For example, Samsung describes that their on-device models are pruned and distilled from a “proprietary” internal model developed by Samsung Research [1]. Google’s Gemini Nano is distilled from larger Gemini models [2].
>
> In the Samsung example, they only deploy the on-device model (no cloud-deployed large models are mentioned in their report), so the internal large models, source models for pruning or teacher models for distillation, are typically not deployed on device. Therefore, their cost should be included for shipping the on-device model. And this is the main scenario we are working on.
>
> In the Google example, the teacher model may be deployed as an API, but it is unlikely to be the source model for pruning because it is usually too large to be a good direct pruning source. For example, the cloud-deployed models are often greater than 20B, but the on-device model is less than 3B. Pruning 20B to 3B often incurs severe performance degradation, so this is typically not a realistic pruning pipeline. In this case, a realistic choice is to either train a 3B model from scratch, or apply the pruning pipeline by training a 7B model and pruning it to 3B (both cases can use distillation with the deployed 20B teacher model). That 7B model is not free; it must be trained, and that cost is exactly what our paper is analyzing.
>
> In summary, existing internal models with medium size (about 7B) are often not deployed on device and are often for pruning purposes, so the cost should be counted. Existing internal models with large size (>20B) are typically too large to serve as direct pruning sources for ≲3B targets, so we still need a medium-size model for pruning and the training cost of the medium-size model should be included.
>
> Under the scenario we are interested in, practitioners do need to consider the training cost of the large model and have the access to the learning rate.
>
> ### **Weakness 2 & Question 2**
>
> > I remain skeptical about whether the proposed method would remain effective for larger LLMs or under higher compression ratios.
>
>
> **About scaling up model sizes**
>
> First of all, our study focuses on on-device models, where the model size is typically less than 3B. Scaling up to larger model sizes is out of the scope of this paper. Please refer to the Common Comments for why 3B is a reasonable size.
>
> Second, within 3B, we have conducted experiments of obtaining a 1.3B model from 2.8B and a 300M model from 600M. Both have shown they reach higher training token efficiency than training the target-size models from scratch.
>
> **About higher compression ratios**
>
> Compression ratio is not a valid metric under our scenario because our paper is not a compression method. That is said, we are NOT interested in the setting where we are given a large model (e.g., 2.8B) and compress it with different compression ratios (e.g., 0.3, 0.5, 0.7) to see how many parameters we can remove while the performance can be maximal preserved. Rather, our paper is a training pipeline to obtain on-device models from scratch.
>
> A similar but different concept under our setting is the enlargement factor. This factor is used to study how large a source model can result in a good on-device model through our pipeline. To answer this question, we have conducted ablation studies of a wide range of factors in Section 5.2 (starting at Line 419). Our experiments show all factors can achieve better performance than training the small model from scratch, and the best factor of practice is between 2x and 4x.
>
> [1] (Samsung Electronics Hosts Samsung Developer Conference Korea 2024, Unveils Its Improved Gen AI Model)[https://news.samsung.com/uk/samsung-electronics-hosts-samsung-developer-conference-korea-2024-unveils-its-improved-gen-ai-model]
>
> [2] Team, Gemini, et al. "Gemini: a family of highly capable multimodal models." arXiv preprint arXiv:2312.11805 (2023).

---

> ### Author Response · Authors · 2025-11-25
>
> Dear reviewer, we'd like to kindly follow up on our responses to your reviews. We would greatly appreciate it if you can let us know whether our responses address your concerns or if any further clarification is needed.

---

### Author Response · Authors · 2025-11-17
**Common Comments**

Dear reviewers, thank you for your time and reviews. We would like to reiterate our motivations/scenarios, the problems under such scenarios, the method scope, and the experiment setup, as we think there are some misunderstandings.

## Motivations/Scenarios
We agree that a common way to obtain small LMs is by pruning or distilling existing open-source large LMs. That scenario exists, and it’s valuable.

**However, we are studying a totally different problem under a totally different scenario that has already clearly existed**: tech companies have sufficient training compute but strict inference budgets; and, due to the license restriction, they are not able to build upon open-source LLMs. Therefore, they have to obtain an on-device model from scratch.

Moreover, training large but serving small is common among tech companies. For example, Samsung describes that their on-device models are pruned and distilled from a “proprietary” internal model developed by Samsung Research [1]. Google’s Gemini Nano is distilled from larger Gemini models [2].

## Problems under the scenario
Under this scenario, we are interested in the problem of how to obtain the best possible on-device (small) models from scratch. One solution is to train the small model with longer tokens, another solution is to utilize pruning. In the application of utilizing pruning, we pose another question: is it worth pretraining a large model even if it is never deployed? This question requires us to compare the full pruning pipeline (including large model pretraining) with small models from scratch.

## Method scope
Our method, IDEA Prune, is considered as a pruning pipeline to produce on-device models from scratch with the help of pruning, NOT a pure pruning technique that aims to compress an existing large model to small models. As a result, our experiments are NOT designed to compare pruned models with the source large models to see how much memory one can save.

## Experiment setup
Our study focuses on on-device models, where the model size is typically less than 3B. Scaling up to larger model sizes is out of the scope of this paper.

Why is 3B a reasonable size? This is the best practice [2, 3] from the perspective of the battery and RAM. **Battery**: Edge devices, such as cellphones, usually have limited and non–on-demand batteries. Battery consumption of an LLM grows linearly with the model size. And since the battery growth has reached a bottleneck for many years [4,5], the size of on-device LLMs will still be limited in the near future. **RAM**: A cellphone typically has 8GB to 12GB RAM. With the OS taking about 4GB and everyday apps usually taking about 2GB, there is only 2GB to 6GB RAM left, which also limits the model to no more than 6B with FP8.

Even though scaling beyond 3B would bring more insights and interesting observations, it would be too expensive to conduct experiments to comprehensively compare pruning pipelines and training-from-scratch baselines. If the experiment of training a 7B model with 2T tokens were conducted, an estimate cost would be 38 days and $1M with 16*8 H100 GPUs.

## Reference

[1] (Samsung Electronics Hosts Samsung Developer Conference Korea 2024, Unveils Its Improved Gen AI Model)[https://news.samsung.com/uk/samsung-electronics-hosts-samsung-developer-conference-korea-2024-unveils-its-improved-gen-ai-model]

[2] Team, Gemini, et al. "Gemini: a family of highly capable multimodal models." arXiv preprint arXiv:2312.11805 (2023).

[3] Liu, Zechun, et al. "Mobilellm: Optimizing sub-billion parameter language models for on-device use cases." Forty-first International Conference on Machine Learning. 2024.

[4] Jennifer Quartararo. “Smartphone Battery Life is Getting Worse.” https://www.electrochem.org/ecsnews/smartphone-battery-life-is-getting-worse?

[5] Chaim Gartenberg. “Why are phones so good but batteries are still so terrible?” https://www.theverge.com/circuitbreaker/2019/6/26/18700773/phone-battery-life-terrible-moores-law-why-technology?

---

### Meta-Review · Area_Chair_pBXJ · 2026-01-09

**Summary:**

This paper received 4 reviews. The reviewers (score/confidence) are: `Q9bX (4/4), an3P (4/4), U8L8 (2/3), aXpd (6/3)`

Their major concerns:

- Motivation: Reviewers argue that practitioners rarely pretrain a large model purely for pruning purposes and instead rely on open-source pretrained LLMs (e.g., LLaMA-3, Qwen-3).  The paper's setting may be "unrealistic".

- Methodology:
  - `Q9bX (4/4)`: the method only focuses on FFN pruning, failing to cover mainstream structured pruning strategies for LLMs such as MHA pruning and layer pruning.
  - `an3P (4/4)`: the core components of IDEA Prune (iterative pruning, sensitivity-based scoring, cosine annealing schedule) are well-established in prior work, and the contribution is only a recombination of existing techniques without fundamentally new algorithms.

- Experiments:
  - `Q9bX (4/4)` and `U8L8 (2/3)`: the experimental setting for only evaluating a single scenario (2.8B → 1.3B), lacking validation on larger models (e.g., 7B/13B) and higher compression ratios.
  - `Q9bX (4/4)` : the paper does not provide analysis of practical inference-time speedup and memory footprint reduction, weakening the demonstration of efficiency benefits.
  - `an3P (4/4)`: hyperparameters (e.g., 50%/50% token budget split) were not tuned for peak performance, reducing the persuasiveness of the experimental results.

**Reviewer Concerns:**

The paper has 3 negative reviewers: `Q9bX (4/4), an3P (4/4), U8L8 (2/3)`:

- Q9bX explicitly stated he/she will maintain the score. The concerns are outstanding.

- an3P did not respond in the rebuttal. He/she was concerned that the paper is a combination of existing techniques. The authors clarified this but did not resolve this concern fully, based on my best evaluation. The proposed techniques are indeed not very new.

- U8L8 did not respond in the rebuttal. He/she was concerned with the "unrealistic" setting and insufficient experiments. These are not resolved during rebuttal.

Besides, as noted by the authors, many reviewers questioned the motivation of this paper - "unrealistic". I also had this concern. Although the authors' summary makes sense to some extent (they said "Our paper is written from the perspective of a company"), this does not change the main concerns. At least, the presented results and the experiment scales can hardly support a claimed "company"-perspetive paper.

**In summary, all the negative reviewers' concerns are outstanding.**

**Reviewer Scores:**

- Q9bX explicitly stated he/she will maintain the score.

- an3P and U8L8 probably will maintain the score too.

---

### Decision · Program_Chairs · 2026-01-26

Reject